



# RainForest: A random forest algorithm for quantitative precipitation estimation over Swizerland

Daniel Wolfensberger[1,2], Marco Gabella[2], Marco Boscacci[2], Urs Germann[2], and Alexis Berne[1]

[1]LTE, Ecole polytechnique fédérale de Lausanne (EPFL), Lausanne, Switzerland
[2]MeteoSwiss, via ai Monti 146, Locarno, Switzerland

*Correspondence to:* EPFL ENAC SIE-IEE-LTE , GR C2 564, Station 2, CH-1015 Lausanne, Switzerland, E-mail: alexis.berne@epfl.ch

**Abstract.**

Quantitative precipitation estimation (QPE) is a difficult task, particularly in complex topography, and requires the adjustment of empirical relations between radar observables and precipitation quantities, as well as methods to transform observations aloft to estimations at the ground level. In this work, we tackle this classical problem with a new twist, by training a random forest (RF) regression to learn a QPE model directly from a large database comprising four years of combined gauge and polarimetric radar observations. This algorithm is carefully fine-tuned by optimizing its hyper-parameters and then compared with MeteoSwiss' current operational non-polarimetric QPE method. The evaluation shows that the RF algorithm is able to significantly reduce the error and the bias of the predicted precipitation intensities, especially for large and solid/mixed precipitation. In weak precipitation, however, and despite a-posteriori bias correction, the RF method has a tendency to overestimate. The trained RF is then adapted to run in a quasi-operational setup providing 5 minute QPE estimates on a Cartesian grid, using a simple temporal disaggregation scheme. A series of six case-studies reveal that the RF method creates realistic precipitation fields, with no visible radar artifacts, that appear less smooth then the original non-polarimetric QPE, and offers an improved performance for five out of six events.

## 1 Introduction

Quantitative precipitation estimation (QPE) is well known to be difficult in orographically complex regions such as the Alps (Houze, 2012; Gabella et al., 2017), due to intricate interactions between the terrain and the precipitation and to a large amount of precipitation falling in solid phase. Still providing an accurate estimate in these regions remains particularly important, because the large precipitation amounts in these regions provide essential water resources. Additionally, the hydrological damages can be severe in steep terrain (e.g. landslides, debris flows) which requires fast and accurate warning systems. The most direct and accurate observations of precipitation intensities are obtained using networks of calibrated and well-maintained raingauges at the ground. Though these measurements are used as a reference, they suffer from inaccuracies in strong wind, especially for solid precipitation (Kochendorfer et al., 2017; Buisán et al., 2017), and provide only a very partial sampling of the precipitation system (Kitchen and Blackall,



1992). Hence, especially for flash-flood and debris-flow alerts as well as hydrological applications at the catchment scale, these measurements need to be complemented with areal measurements, typically provided by weather radars. Unfortunately, radar measurements are particularly prone to errors and uncertainties in mountainous regions, due to the partial or total beam-blocking by the orography, which restricts the observations to higher altitudes (e.g.,

Gabella and Perona, 1998; Germann et al., 2006; Anagnostou et al., 2010). In addition, QPE in solid precipitation is also much more difficult due to the vast heterogeneity of solid hydrometeors and the complex relation between radar observables and intensity (Fujiyoshi et al., 1990; Zrnic and Ryzhkov, 1999).

Traditionally, QPE has involved adjusting relations between polarimetric radar observables and precipitation intensities based either on *in-situ* observations by disdrometers (e.g., Joss et al., 1998; Chapon et al., 2008; Tokay

et al., 2009) or by matching the resulting precipitation estimates with gauge observations (Mapiam et al., 2014). While the first approach has the advantage of being physics-based, there is no guarantee that the derived relations are still valid at larger scales (Verrier et al., 2013), and as such it often requires additional bias correction with gauges as reference (Morin and Gabella, 2007). The second approach has the disadvantage of relying too much on potentially flawed gauge observations, and is often based only on a limited number of precipitation events. Though

power-laws are traditionally used as mathematical models to relate radar observables to precipitation quantities, some efforts have been made to train artifical neural networks (ANN), which are machine learning models able to represent any mathematical function (Cybenko, 1989). An issue with ANN however is the difficulty to fine-tune them accurately in the presence of noise, which can lead to overfitting and physically unrealistic outcomes.

Currently MeteoSwiss relies on a two-step process to provide the best possible QPE. The first step is a radar-based

real-time QPE, which relies on a unique Z-R relationship to convert radar reflectivity to precipitation aloft. This estimate is then corrected for partial beam shielding and extrapolated to the ground with a dynamical vertical profile of reflectivity (Germann et al., 2009). The second step improves this radar-based estimate by merging it with gauge observations, using a geostatiscal interpolation technique called co-kriging with external drift (Sideris et al., 2014), to provide an hourly QPE estimate, which is then disaggregated to a 5-minute resolution (Barton et al., 2020). Since

the development of the radar-based QPE, the radar network of MeteoSwiss has been updated significantly: it now consists of five dual-polarization, Doppler, C-band radars (Germann et al., 2015). The update to dual-polarization offers wide opportunities, and the rich additional information it provides is already used operationally for the classification of hydrometeors from radar measurements(Besic et al., 2016) and the identification of ground clutter. Dual-polarization brings additional information especially in intense precipitation (Ryzhkov et al., 2005, 2014) and

solid precipitation (Ryzhkov and Zrnic, 1998; Bukovčić et al., 2018).

The goal of this work is to derive a new data-driven radar-based QPE algorithm that provides accurate precipitation estimates in Switzerland's complex topography and takes advantage of the large archive of polarimetric radar data collected over the years by MeteoSwiss's operational radar network. The algorithm should be as direct as possible to avoid the use of *a-posteriori* bias-corrections, and should also provide uncertainty estimates. This algo-





rithm should with time replace the first-step of the QPE estimation, and provide a better input to the gauge-radar merging, which will hopefully also lead to a better final output.

To reach these ends, a non-parametric model for QPE is developed, that does not rely on specific power-laws, but uses random forest (RF) regression to learn a model directly from the data. By feeding it with appropriate input

features, the model is able to natively correct the predictions for bright-band and calibration issues and extrapolate precipitation to the ground level, thus simplifying the overall processing chain. Orellana-Alvear et al. (2019) recently presented promising results with a RF approach in the Andes, although with only a single-polarization X-band radar. However the alternatives to the RF models that are considered in their work are quite simplistic (Marshall-Palmer ZR relation, Marshall and Palmer (1948) and custom fitted power-law), and do not include the typical bright-band

and local bias corrections that are present in operational QPE models In this work go further by considering the full polarimetric radar and ground station network of Switzerland (5 C-band radars and more than 270 ground stations), over the course of four years of observations, and we compare the performance of this model with the operational state-of-the-art QPE products processed at MeteoSwiss.

This article is structured in the following way: Section 2 provides an overview of the database that was used to

train and evaluate the QPE method, Section 3 introduces the random forest regression and the transformation of input data it requires as well as the performance metrics that are used throughout this work. At the end of the section, these metrics are used to evaluate the performance of MeteoSwiss' current QPE products. Section 4 details the overall performance and the optimal configuration of the random forest QPE. Section 5 completes the previous section by explaining how the algorithm was adapted to a quasi-operational mode where it provides 2D maps of

precipitation every 5 minutes. The performance of the new QPE algorithm is then assessed on a case study of six precipitation events. Finally Section 6 concludes this work and summarizes the main advantages and limitations of the proposed method.

## 2   Collocated radar/gauge database

Training a machine learning algorithm requires large amounts of data in a homogeneous format. Even though the

present archives of MeteoSwiss contain vast amounts of data covering decades of measurements, these data have different spatial and temporal resolutions (from point measurements to large area COSMO runs), are sometimes temporally inhomogeneous and are stored in different file formats. Thus an important effort has been invested in the creation of a homogenized dataset that can be used to train any type of machine learning model with the main objective of precipitation estimation but also allowing for other potential uses (e.g. verification of operational

products and correction of bias). Note however that only the data that is explicitly used in the present QPE study will be detailed in this section.

Most MeteoSwiss operational products are estimated over a Cartesian grid of 1 km$^2$ (in the Swiss LV03 coordinate system) at a temporal resolution of 2 to 5 minutes. Archives from the operational COSMO numerical weather



|  | **Original resolution** | | **Database resolution** | |
|---|---|---|---|---|
|  | spatial | temporal | spatial | temporal |
| Radar | $1° \times 500$ m $\times 20$ elevations | 5 min | $1$ km$^2$ $\times 20$ elevations | 10 min |
| COSMO (model) | $\approx 1$ km$^2$ $\times 60$ vert. levels | 1 h | $1$ km$^2$ $\times 20$ elevations | 10 min |
| Operational products | $1$ km$^2$ | 5 min | $1$ km$^2$ | 10 min |
| Synoptic stations | Point | 10 min | Point | 10 min |

**Table 1.** Native and transformed spatial and temporal resolution of the products included in the gauge-radar database.

prediction model (Seifert et al., 2011; Doms et al., 2011; Baldauf et al., 2011; Wolfensberger and Berne, 2018) are available over Switzerland every hour[1] on a 3D irregular grid (e.g., Gal-Chen and Somerville, 1975; Wolfensberger and Berne, 2018). Polarimetric radar data is available every 5 minutes on a polar grid. Finally, the synoptic weather station data has a temporal resolution of 10 minutes. To accommodate these differences, the reference temporal

resolution of the database is 10 minutes and the reference spatial resolution is 1 km$^2$ for spatial data (Cartesian products and polar radar data).

For Cartesian and polar data, the aggregation to 10 min resolution is done by simple averaging. For quantities expressed in decibels such as radar reflectivity, the averaging is done on linear quantities and the average is converted to decibels. For radar data, three methods for the spatial aggregation to a 1 km$^2$ pixel have been used: *mean*, where

the average of all observables that fall within a given 1 km$^2$ pixel is taken (with the same consideration as above for decibel quantities), *max*, where only data at the polar gate with maximum $Z_H$ (within square km) is taken and *min*, where only the data at the polar gate with minimum $Z_H$ (within square km) is taken. For COSMO data only the *mean* aggregation method is used in space, whereas in time a linear interpolation between hourly outputs is made to get to 10 min temporal resolution.

For radar data and Cartesian products, the extraction is performed separately for a $3 \times 3$ pixels neighborhood around the center pixel, in which the synoptic station is located. The data corresponding to the different neighbours are then stored as separate columns in the database.

The database covers four years of measurements from January 2016 to December 2019 for the 5 Swiss radars. To avoid populating the entire database with zeros, at a given station, only the 10 minute timesteps that fall within

an hour where the rain gauge recorded at least 0.1 mm of precipitation[2] were included. Note that even if there are no dry hours in the database, at the 10 minute resolution, the proportion of observed zero precipitation intensities is still 30 %. The database consists of around 3.3 million observations at the ground, every row corresponding to a different combination of 10 minute timestep and station; and 18 million radar observations aloft (for the station

---

[1]the temporal resolution of the COSMO model is much higher but since the data amount is huge, 3D archives are kept only at hourly resolution

[2]which corresponds to one tip of a tipping bucket rain gauge and is the maximum resolution of all rain gauge



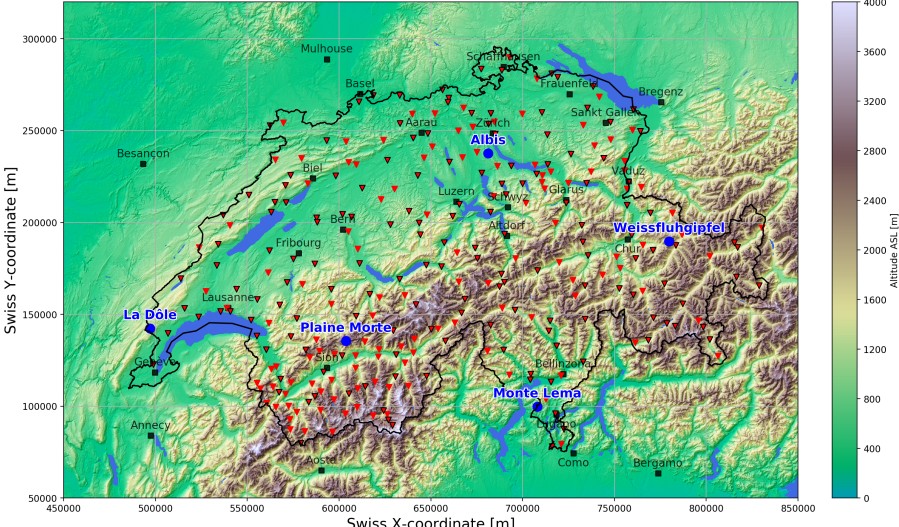

**Figure 1.** Topography of Switzerland with the five Swiss operational radars (blue circles), the 160 synoptic weather stations (red triangles with black border) and the 128 rain gauges (red triangles without borders). Major cities are indicated with black squares. This map is based on a digital elevation model provided by SwissTopo.

pixel only, the number of observation for neighbour pixels is similar). Aggregated to hourly resolution this represents a total of around 550,000 station-hours at the ground (hourly observation at a given station).

## 2.1 Synoptic weather station data

Synoptic weather data comes from the SwissMetNet (SMN, Suter et al. (2006)) observation network which regroups more than 288 stations from which 160 are synoptic weather stations and 128 are rain gauges only, which only record the precipitated amounts. Note that the area of Switzerland is around 41'000 km$^2$, and the average distance between two stations is 11 km. These stations provide observations every 10 minutes. Two station observations were used in this study: the precipitation amount over 10 minutes measured at a height of 1.5 meters and the temperature at a height of 2 meters. In some stations, precipitation measurements are performed with a tipping bucket Lambrecht rain gauge (types 1518 H3 et 15188), but in most stations an Ott Pluvio$^2$ weighing rain gauge is used instead. All rain gauges are heated to melt solid precipitation, but are not shielded from the wind. Temperature measurements are performed with a MeteoLabor Thygan instrument.

Figure 2 shows the distribution of hourly precipitation observations for the entire database. It can be clearly seen that the distribution is strongly right-skewed, with a vast majority of small intensities and very few but intense extremes. Note that roughly half of the ground observations in the database comes from weather stations and the other half from rain gauges, where no information about air temperature is available. From the weather station observations, only 16 % correspond to temperatures below 0°C.





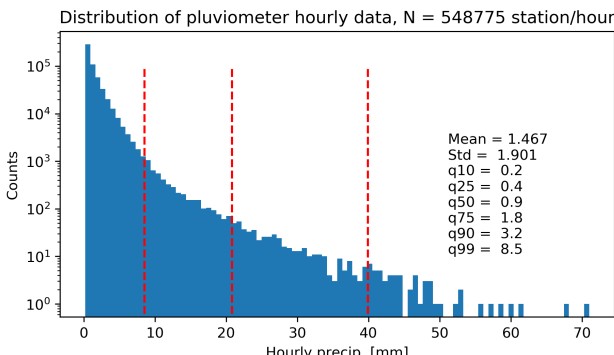

**Figure 2.** Distribution and statistical indicators of hourly observed precipitation intensities for the entire database (January 2016 to December 2019). The three vertical dashed red bars indicate the percentiles 99, 99.9 and 99.99.

## 2.2 Radar and COSMO data

The Swiss radar network consists of five polarimetric C-band radars which perform Plane Position Indicator (PPI) scans at 20 different elevation angles[3] (Germann et al., 2006), using an interleaved scanning strategy. The polar data used in this study consists of the final quality checked measurements, corrected for ground clutter and calibration and aggregated to a radial resolution of 500 m (over 6 consecutive range gates). In addition to radar observations, the temperature from the COSMO numerical weather prediction model has been interpolated to the radar grid using nearest-neighbour interpolation. The radar and COSMO variables that have been used in this study are listed in Table 2.

## 2.3 MeteoSwiss Cartesian reference products

Two types of MeteoSwiss Cartesian products have been used in this study.

**RZC**

RZC is the standard operational purely radar QPE product of MeteoSwiss (Germann et al., 2006; Gabella et al., 2017). It provides 2D maps of precipitation intensities in mm h$^{-1}$ equivalent of liquid water every 5 minutes.The algorithm starts by estimating the precipitation intensity at every radar gate from the reflectivity with the power-law $Z = 316R^{1.5}$ (Joss et al., 1998), where $Z$ (linear reflectivity) is in units of mm$^6$m$^{-3}$ and $R$ (precipitation intensity) in mm h$^{-1}$. Prior to this transformation, gates with low visibility VIS (VIS $\leq$ 37%) are discarded. . The values are corrected for partial beam shielding by applying a multiplicative correction of 100 / VIS (in %). To account for growth and decay of precipitation with altitude, a correction with a dynamic vertical profile of reflectivity (Germann and Joss, 2002) is then applied to every $R$ value aloft.The $R$ values aloft are integrated to the ground using a weighted

---

[3]-0.2, 0.4, 1.0, 1.6, 2.5, 3.5, 4.5, 5.5, 6.5, 7.5, 8.5, 9.5, 11.0, 13.0, 16.0, 20.0, 25.0, 30.0, 35.0 and 40.0°



| Name | Description | Units |
|---|---|---|
| Visibility | Static visibility of a given radar volume obtained statically with a DEM and a radar refraction model | % (0% = total blockage) |
| $Z_h$ | Reflectivity factor at horizontal polarization corrected for visibility using a factor of 100 / visibility (in %) | $mm^6\ m^{-3}$ |
| $Z_v$ | Reflectivity factor at vertical polarization corrected for visibility as above | $mm^6\ m^{-3}$ |
| $K_{dp}$ | Specific differential phase shift upon propagation obtained with the method detailed in Appendix A. | $°\ km^{-1}$ |
| $R_{vel}$ | Mean Doppler (radial) velocity | $m\ s^{-1}$ |
| $S_w$ | Spectral width (standard deviation of Doppler velocities within a radar resolution volume) | $m\ s^{-1}$ |
| $A_h$ | Specific attenuation at horizontal polarization | $dB\ km^{-1}$ |
| $N_h$ | Estimated noise level at every gate | dBm |
| T | Temperature from the COSMO model interpolated to the radar polar coordinates | $°\ C$ |

**Table 2.** List of radar and COSMO variables used aloft the synoptic stations.

sum, linearly related to the visibility and exponentially related to the height of observations: $w(h) = \exp(-0.3h) \cdot \frac{VIS}{100}$, where $h$ is the height above ground of the observation in meters. Obviously the negative factor in the exponential implies that observations closer to the ground have a larger weight. These weighted averages are then resampled to the Cartesian 1 $km^2$ grid. Finally a multiplicative local bias correction is applied at every Cartesian pixel to obtain

the final $R$ estimated at the ground (see Germann et al. (2006) in particular "Experiment" in Table 2 (p. 1684), Figure 8 (p. 1686) and Section 5).

**CPC**

CPC is a combined gauge-radar QPE product developed by Sideris et al. (2014). The merging is performed with a geostatical method called co-kriging with external drift, in which the spatial dependence of radar and gauge

observations are fitted dynamically with an exponential law. The gauge data is then interpolated in space and time (co-kriging) to the Cartesian grid as a primary variable using the radar data as a trend (drift). This method only yields an hourly estimate but a recent algorithm by Barton et al. (2020) is used operationally to produce 5 minute CPC estimates, by disaggregating hourly CPC estimates with hourly fractions of 5 minute RZC estimates. Also note that at every gauge in Switzerland an hourly cross-validation product called CPC.CV is computed using a

leave-one-out strategy (the gauge for which the CPC performance is assessed is not used in the algorithm).





# 3  QPE computation

## 3.1  Choice of a regression method

Thanks to this large database of collocated gauge and radar observations, a QPE model can be trained and used for
further prediction on new data, providing a 2D Cartesian estimate on the same grid as the current QPE product
(Section 2.3).

   To be used in an operational context, the QPE method must be fast (real-time use) and robust in the case of
faulty radar measurements, both during training and subsequent prediction of new values. Obviously, it should take
benefit from polarimetric information which is not used in the current RZC method. Moreover, unlike the current
method it should provide an unbiased estimate, that does not require additional local corrections. Three machine
learning regression methods were considered: artificial neural networks (ANN), gradient boosting (GB) and random
forests (RF). The advantage of RF is the simplicity of the hyperparameter tuning and the inherent parallelization of
the training and predicting. RF are not able to extrapolate, meaning that the input dataset has to be representative
of all cases that can be encountered in nature. ANN are powerful and easy to parallelize but require careful tuning
and can easily overfit in case of noise, leading to unphysical predictions. GB can be extremely powerful and do not
extrapolate but are harder to parallelize and to tune. Preliminary tests showed that without extensive fine-tuning
the three methods provide relatively similar performance. Due to its numerous advantages it was thus decided to
use only Random Forest regression.

## 3.2  Random Forest regression

Random Forests (Breiman, 2001) are an ensemble learning methodology, where the outcomes of a number of trained
weak learners (in this case decision trees) are combined with a voting scheme to yield a boosted estimate with a
better performance. This is inspired by the wisdom of the crowd process, where a collective heterogeneous group of
individuals is better at analyzing and solving a complex problem than single individuals, even if they are experts. To
guarantee the heterogeneity of the weak-learners, RF includes bootstrap resampling and random feature selection.
Let us assume that the input dataset has dimension $NM$, where $N$ is the number of samples and $M$ the number of
input features. For each tree in the forest, a new training set with $N$ samples is created using bootstrap sampling
(random selection of samples with replacement). For each training set a new decision tree is grown using the CART
method (Breiman et al., 1984). Every time a new split has to be made at a given node of the tree, only a number
$m$ of features ($m < M$, typically $m = \sqrt{M}$), are randomly selected. This process, which is trivial to parallelize is
repeated until $t$ trees are grown, giving a random forest. In the case of random forest regression, the final prediction
is simply the average of all outcomes of the individual trees of the forest. The hyperparameters of the random forest
regression model, which need to be fine-tuned with cross-validation are:

   − the number of trees $t$ in the forest





- the maximum depth $d$ of the individual trees (i.e. how many times a split is made). Trees that are too shallow will be too much biased, whereas trees that are too deep will be overfitted

- The number of features $m$ randomly considered when splitting a decision tree

- The minimum number of samples in a node to split it

- The minimum number of samples in a leaf (child node) to accept a given split

Because RF regression uses the average of the tree predictions, they tend to underestimate extreme values and overestimate small values (Zhang and Lu, 2012). Even if it is very rare and does contribute only marginally to the total precipitation amounts, extreme precipitation is a key part of QPE, since it causes the largest impact on landscapes, ecosystems and human activities. Consequently, to allow RF to better represent large values, the two

last parameters (minimum number of samples in a node and in a leaf for a split) have been set to 2 and 1, which is also the default in the *scikit-learn*, (Pedregosa et al., 2011) machine-learning library that was used to train the RF algorithm. This implies that the splitting procedure is not affected by the size of the node and the generated leaves.

Moreover, in order to further mitigate the inherent bias of RF, three types of *a-posteriori* bias correction (BC) methods were compared.

**BC_raw** a polynomial regression of predictions versus observations (from gauge) on the training dataset is performed and this fit is then used to correct new RF predictions.

**BC_cdf** a polynomial regression of *sorted* predictions versus *sorted* observations on the training dataset is performed, this fit is then used to correct new RF predictions. This can be seen as a form of histogram matching, since it maps the cumulative density function (CDF) of predictions to the CDF of observations.

**BC_cdf_spline** same as *BC_cdf* but a cubic spline is used instead of a simple linear regression.

Figure 3 shows an example of predicted values versus observations on the training fraction and the first order bias-correction methods that were fitted to the data. Comparisons with the 1:1 line show that high intensities are generally underestimated. The bias-correction methods apply a factor to every new prediction that should bring them closer to the 1:1 line. Note that the relative performance of these BC methods needs to be assessed on an

independent test dataset.

### 3.3 Transformation of radar data

In the database a column of radar observations is available aloft over every station. Reference precipitation observations are however only available at the ground. Machine learning methods require consistent dimensions of input features and response (observations). Therefore, radar data needs to be aggregated to the ground level. In our model,

taking example on the current RZC QPE, the radar data is aggregated to the ground using a similar weighted sum:





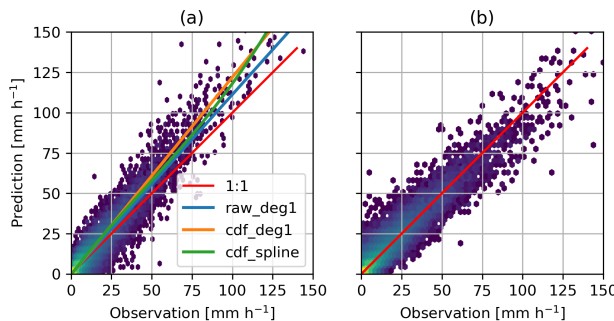

**Figure 3.** Panel (a) shows ab example of raw uncorrected predictions (as a density plot) versus observations on the training fraction. The blue, orange and green lines are the corresponding fitted bias-correction models, which are estimate the observations as a function of raw predicted values. Panel (b) shows the raw predictions corrected with the BC_cdf_spline method, which brings them much closer to the 1:1 line.

$$w(h) = \exp(-\beta h) \cdot \frac{\text{VIS}}{100} \qquad (1)$$

where the $\beta$ parameter indicates the slope of this exponential, and was fine-tuned with cross-validation alongside the other RF hyperparameters (Section 3.2).

This transformation allows to derive five additional variables: $\text{Frac}_{\text{radar\_r}}$, which the fraction of observations aloft that come from radar $r$ ($r$ being one of the five operational radars). This fraction is weighted in the same exponential way, meaning that for a given radar, the presence of observations at low altitudes gives a larger increase in the fraction. Note that since these variables are all related to the others, they will be grouped together under the general term $\text{Frac}_{\text{radar}}$.

### 3.4 Performance metrics

In order to assess the performance of the QPE method and to compare it with the current RZC algorithm, pertinent performance metrics are required. A single metric is usually not sufficient to represent the error structure, hence in this work we will use four different complementary metrics. Let us use the notation $Y$ for the response variable (observed precipitation intensity) and $\hat{Y}$ for the QPE estimation.

**RMSE** The root mean square error in units of mm h$^{-1}$

$$\text{RMSE} = \sqrt{\frac{1}{N} \sum_{i=1}^{N} (Y_i - \hat{Y}_i)^2} = \sqrt{\text{ME}^2 + \text{STDE}^2} \qquad (2)$$

where ME is the mean linear error (bias) and STDE the standard deviation of the errors. RMSE, because of the use of an exponent two is quite sensitive to large deviations, occuring for high precipitation rates values.





**scatter** Weighted interquantile (16 - 84 %) of relative bias in units of dB (Germann et al., 2006)

$$\text{scatter} = 0.5 \cdot (Qw_{84}(\epsilon_{\text{dB}}) - Qw_{16}(\epsilon_{\text{dB}})), \tag{3}$$

where

$$\epsilon_{\text{dB}} = 10 \log\left(\frac{Y_i}{\hat{Y}_i}\right), \quad i = 1, .., N$$

and $Qw$ is a weighted quantile (Edgeworth, 1888), where the weights $w$ are related to the observed precipitation intensity:

$$w = \frac{\hat{Y}_i}{\sum_{i=1}^{N} \hat{Y}_i}$$

**logBias** The relative bias in units of dB

$$\text{logBias} = 10 \log\left(\frac{\sum_{i=0}^{N} Y_i}{\sum_{i=0}^{N} \hat{Y}_i}\right) \tag{4}$$

**ED** The energy distance, which is a unitless measure of the statistical distance between two distributions (Rizzo and Székely, 2016).

$$\text{ED}(Y, \hat{Y}) = \sqrt{2\ E||Y - \hat{Y}|| - E||Y - Y'|| - E||\hat{Y} - \hat{Y}'||} \geq 0 \tag{5}$$

The prime symbol indicates the difference between pairs of successive values and the norm $||$ is the standard euclidean norm.

The two first metrics are estimated of the error of the QPE model, the RMSE is a measure of the additive error and is more sensitive to extreme values, whereas the scatter is a robust measure of the relative error, since it ignores the tails of the distribution. The third metric is a measure of the relative bias of the QPE model, expressed in logarithmic scale. The last metric measures the match of the predicted precipitation distribution with the observed values. As such it does not indicate if a single predicted precipitation value is correct but only that the global population of predicted values is representative of what is observed in nature. As in (Sideris et al., 2014; Speirs et al., 2017; Panziera et al., 2017) the performance metrics will be mostly evaluated at hourly resolution (aggregation of 6 consecutive 10-minute timesteps), because of the limited representativeness of the gauge data (due mostly to the spatial underesampling of the gauge but also to wind effects and limited accuracy of the instrument). This also avoids numerical issues in the logarithmic scores (logBias and scatter) since all hours in the database are rainy, whereas some 10 minute timesteps are dry.

### 3.5 Performance of reference products

Figures 4 shows the scatter-plots of observed precipitation versus reference products (CPC, CPC.CV and RZC) for all observations and for observations with $T < 2°C$, which might correspond to solid/mixed precipitation. Clearly,





CPC delivers by far the best performance for all evaluation metrics, except logBias which shows the tendency of CPC to underestimate strong precipitation, in particular in snow, a consequence of the smoothing caused by the kriging algorithm. However, since CPC is taking into account the observed gauge measurement, it is not a fair comparison. We will thus restrict the evaluation to the CPC.CV and RZC products. Clearly, RZC has a relatively
large overall RMSE, especially for larger intensities, it is however relatively unbiased and has a low ED, indicating that it provides realistic, although sometimes inaccurate precipitation estimates. It tends however to underestimate quite strongly solid precipitation intensities. CPC.CV provides a systematically better performance than RZC, and the improvement is particularly clear in solid/mixed precipitation. Note that decreasing the temperature threshold from 2 to 0° decreases the performance on all scores by 10 to 30%, but it affects all models in a similar way and as
such does not change the general conclusions.

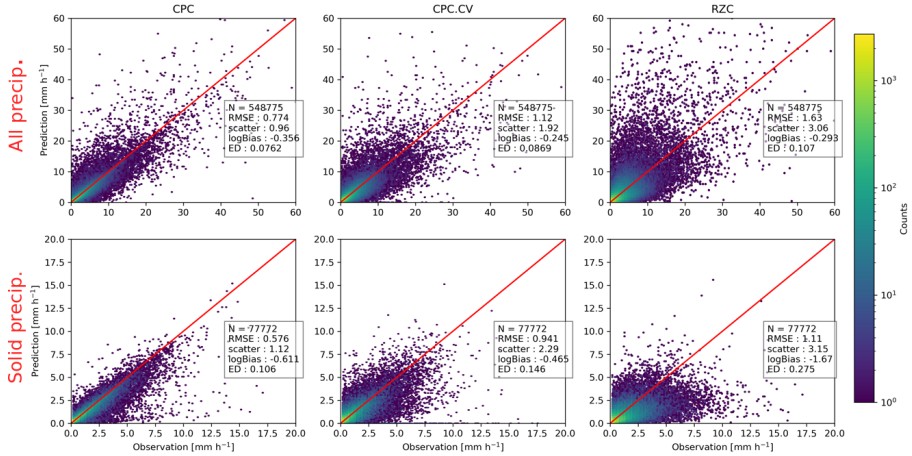

**Figure 4.** Scatter-plots and performance scores of the reference products CPC, CPC.CV and RZC for all available observations (upper row) and for observations with $T < 2°C$ (lower row). The 1:1 line is shown in red. The colorbar gives the counts in logarithmic scale.

### 3.6   Filtering of input data

To avoid including spurious data in the training and validation procedure of the random forest. The following data was excluded from the study:

1. Data from the 'TIT','GSB','GRH','PIL','SAE' and 'AUB' stations, where radar agreement has always been
poor, because of poor radar visibility (very complex topography) and/or suboptimal rain gauge location (wind-induced undercatching). With the exception of the last one (located at 1100 m but with poor radar visibility), all of these stations are located above 1900 meters.

2. Data where $Z_h$ aggregated to the ground is larger than 20 dBZ and the gauge measures no precipitation


3. Data where $Z_h$ aggregated to the ground is smaller than 5 dBZ and the gauge measures more than 0.5 mm h$^{-1}$ equivalent.

The two last constraints reduce the effect of strong advection which leads to a decorrelation between gauge and radar observations, due to temporal and spatial shifts of the precipitation field. These three criteria lead to 6.5 % of the data being filtering out (from which fraction condition 1 represents 20%, condition 2, 35% and condition 3, 45%).

## 4    Fitting of a QPE model and results

### 4.1    Choice of input features

To assess the relative importance of all available input variables (Table 2) aggregated to the ground as in Section 3.3, and choose the most pertinent ones, an approach from Han et al. (2016) has been adapted to regression. Assuming as before that $M$ is the number of available input features, the method is described in Algorithm 1.

---

**Algorithm 1** Mean increase in RMSE

1: **for** $k$ in $K$-fold cross-validation **do**
2:     Split dataset randomly into test and train fractions
3:     Train random forest regressor on train fraction using all input variables
4:     Compute RMSE on test fraction : RMSE$_{\text{ref}}$
5:
6:     **for** $j \in \{1,...,M\}$ **do**
7:         Randomly shuffle input feature $j$ on test fraction, keeping the other features untouched
8:         Compute RMSE on shuffled test fraction : RMSE$_{\text{shuffled}}^{(j)}$
9:         Compute increase in RMSE: score$^{(j)}[k] = \frac{RMSE_{\text{shuffled}}^{(j)} - \text{RMSE}_{\text{ref}}}{\text{RMSE}_{\text{ref}}}$
10:     **end for**
11: **end for**
12: **for** $j \in \{1,...,M\}$ **do**
13:     Compute mean increase in RMSE : $\overline{\text{score}^{(j)}} = \frac{1}{K}\sum_{i=1}^{K}\text{score}^{(j)}[k]$
14: **end for**

---

In this study, cross-validation is always done by separating all precipitation events in the dataset and by randomly attributing these events to either the train or test fraction. We define precipitation events as a continuous period of precipitation observations with less than 12 hours of time interval between every observation. In other words, two successive precipitation events are separated by a dry period of at least 12 hours in between them. Precipitation events always cover full hours (from o'clock to o'clock). This method has the double advantage of increasing the independence between test and train fraction and avoids including partial hours into the test and train fractions,





which is an issue when the evaluation metrics are computed at hourly aggregation. In addition, K in the K-fold cross-validation is always set to 5 (5 iterations).

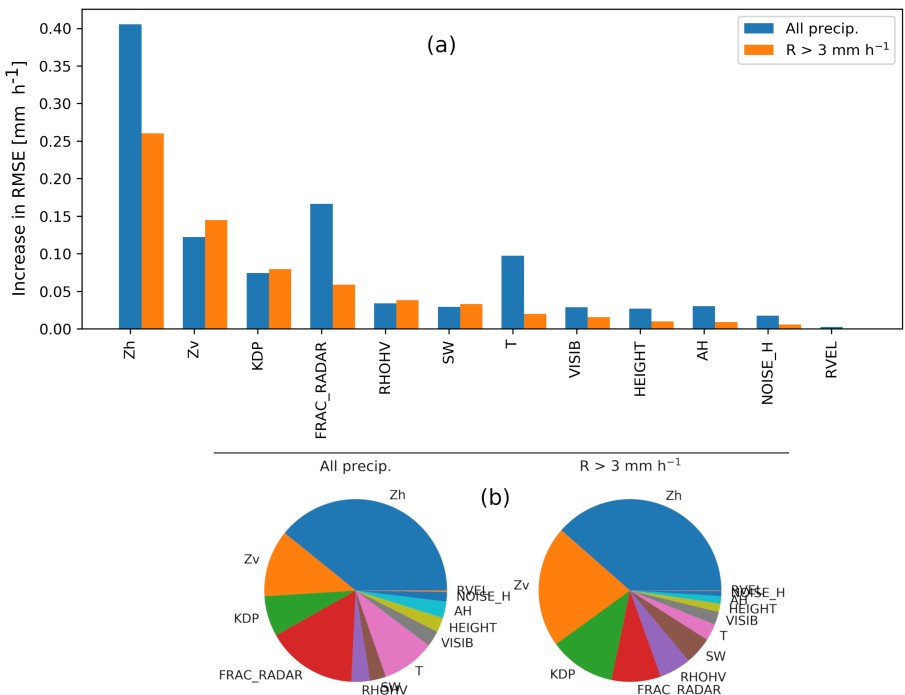

**Figure 5.** (a) Mean decrease in RMSE obtained with algorithm 1 for all available features, separated for all precipitation and for $R > 3$ mm h$^{-1}$, (b) relative importance of these RMSE decreases (normalized by the sum of all decreases) .

The results of Algorithm 1 are shown in Figure 5, separately for all observations and for high precipitation intensities. As expected the horizontal reflectivity $Z_h$ has by far the highest importance, and is followed by the
5   polarimetric variables $Z_v$, $K_{dp}$. The fraction of every radar also has a large importance, this can be explained by two factors. First of all the individual radars are not perfectly homogeneous and differ slightly in calibration, this is taken into account in the RZC product by global biases applied separately to all radar observations. Secondly, this is a way for the regressor to account for the spatial precipitation structure over Switzerland, for example Alpine regions with relatively poor visibility, where precipitation tends to be underestimated by the RZC product are characterized by
10  low radar fractions from the three lower radars (Albis, La Dôle and Monte Lema). Surprisingly the spectral width $S_w$ seems to play a relatively large role, which can be due to the fact that it is an indicator of convection (Hooper et al., 2005; Rao et al., 2010), which leads to different relations between precipitation and radar observables. Note also that the relatively low importance of the height (in fact this is the average weighted height of the observations





aloft) and the visibility is likely explained by the fact that these variables are already included in the exponential weighting (Section 3.3 and, for the visibility, in the correction of $Z_h$ and $Z_v$ (Table 2). Finally $A_h$, $\text{Noise}_H$ and $R_{\text{vel}}$ have low importance on average. For $A_h$, this is somehow in contradiction to Ryzhkov et al. (2014) which give very promising results for QPE. However their study was at X-band and requires ray to ray fine-tuning of the power-law

parameters of the ZPHI method, whereas we used only constant default parameters. There is work in progress to improve the estimation of $A_h$ at MeteoSwiss, but it is a tedious task in complex topography and is thus out of the scope of this work. One should not forget however that even variables with low average importance might in some particular cases be very informative, for example attenuation in the case of a gauge behind a strong thunderstorm.

It is also interesting to notice that for larger intensities, the pie-charts in Figure 5 show a relatively larger

importance $S_w$ (which is an indicator of convection) and of polarimetric variables $Z_v$, $K_{dp}$, $\rho_{hv}$ and a clear smaller importance of the radar fractions, since discrepancies between radars become weaker for strong signals and the temperature, since high intensities are mostly related to strong convection, with liquid precipitation at high altitude.

In the final choice of input variables, it was decided to include all 13 features[4] displayed in Figure 5 with the exception of $A_h$, $\text{Noise}_H$, $R_{\text{vel}}$, because of their low importance and additional computational cost (for $A_h$). This

model will be refered to as RF_dualpol. For sake of comparison and to evaluate the possible performance in case of a failure in the vertical polarization channel, one additional model will be tested: RF_hpol, where only horizontal polarization is available, and $\rho_{hv}$, $K_{dp}$ and $Z_v$ are not include.

## 4.2    Optimization of hyper-parameters

Once the input features have been specified, a grid-search method has been used to find the best possible hyper-

parameters. For every combination of hyper-parameters, a 5-fold cross-validation is performed in order to get the average performance metrics. The following hyper-parameters have been tested:

The four performance metrics (Section 3.4) were estimated at hourly aggregation for observed precipitations ranges $\leq 2$ mm h$^{-1}$, $2 - 10$ mm h$^{-1}$ and $\geq 10$ mm h$^{-1}$ and for solid precipitation ($T \leq 2°$C) and averaged over all cross-validation iterations. The combination of hyperparameters providing the best trade-off between performance

for all metrics over all precipitation subsets and computational cost is then found with:

$$\text{idx}_{\text{best}} = \underset{c}{\text{argmin}} \quad 0.3 \cdot \text{Cost}_{\text{computation}}[c] + 0.7 \, \text{Cost}_{\text{performance}}[c] \tag{6}$$

where $c$ is a given combination of all hyperparameters $[t, d, m, \beta, \text{BC}]$. The computational cost is proportional to the number of trees and their maximum depth.

$$\text{Cost}_{\text{computation}}[c] = d \times t \tag{7}$$

---

[4]as stated previously the radar fraction is decomposed into 5 features, one for every radar





| Parameter | Meaning | Tested values |
|---|---|---|
| $t$ | Number of trees in the random forest | 10,15,20,30,40,50 |
| $d$ | The maximum depth (number of nodes) of the individual trees | 10,15,20,30,40 |
| $m$ | The number of features randomly picked when doing a node split | 1, 3, 5, 7, 9, 11, 13 |
| $\beta$ | The exponent in the exponential altitude weighting (Equation 1) | -0.3, 0.5, 0.7, 0.9 |
| BC | The type of bias-correction method (Section 3.2) | No BC, 'BC_cdf' with a fitted polynome of degree 1, 2, 3 , 'BC_raw' with a fitted polynome of degree 1, 2, 3, 'BC_cdf_spline' |

The performance cost is given by

$$\text{Cost}_{\text{performance}}[c] = \sum^{\text{PS}} \frac{1}{6}\text{RMSE}_z^{(PS)}[c] + \frac{1}{6}\text{scatter}_z^{(PS)}[c] + \frac{1}{3}\left|\text{logBias}_z^{(PS)}\right|[c] + \frac{1}{3}\text{ED}_z^{(PS)}[c] \qquad (8)$$

where the suffix $z$ indicates a score standardized over all samples $N$: ($z$-scores):

$$\gamma_z = \frac{\gamma - \overline{\gamma}}{\sigma(\gamma)}$$

Note that the weights of RMSE and scatter are halved with respect to the weights of logBias and ED since they are both a measure of error dispersion. The final best trade-off was found with $t = 15$, $d = 20$, $m = 7$, $\beta = -0.5$ and BC = 'BC_cdf_spline'. A comparison of the effect of the choice of hyperparameters on the RMSE, for all precipitation and for high intensities is shown in Appendix B.

### 4.3   Stability of the model

A good way to diagnose the completeness of the training data and the stability of a machine learning model is to compute a learning curve (Meek et al., 2002; Praz et al., 2017), that shows the performance on the test and train fractions with increasing number of samples used for training. When looking at this learning curve in Figure 6, one can conclude that the size of the database seems sufficient to train the model, as the test error reaches a plateau for a high number of samples and does not decrease significantly with the number of samples. Note that for random

forest regression the train error and its variability tend to be very small, since when considering a large maximum depth of the individual trees (large $d$), the model is generally able to (almost) perfectly render the response variable on the training set. Because of this, the interpretability of the train error in this case is very limited, and it is hence not displayed.





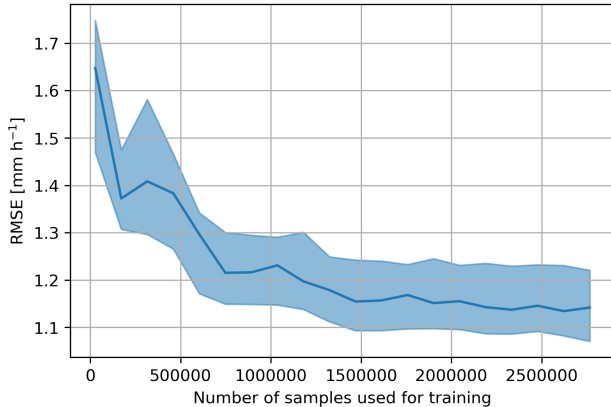

**Figure 6.** Learning curve of the RF_dualpol model. It shows the RMSE of the test fraction at hourly scale as a function of the number of samples used for training. The curve was obtained with 10 iterations of 5-fold cross-validation. The coloured areas correspond to the Q25-Q75 interquantile and the solid line to the mean.

Another hypothesis of this QPE model is that the training dataset is consistent through time and there is no major change in the input features that is caused by non-natural phenomena (such as hardware modifications or additional beam-shielding caused for example by a new construction in the vicinity of a radar). One way to verify this is to look for a trend in the time series of daily cross-validation errors. To this end, the approach of Cleveland

et al. (1990) was used to decompose the time series into daily fluctuations, seasonal trend and long-term trend. The results are shown in Figure 7. It appears that there is indeed no long-term trend in the cross-validation error and besides the obvious seasonal trend (larger intensities in summer), there is no tendency of the model to perform better or worse over some periods of time. As such, if the current radar setup in Switzerland is conserved (as planned), the model should only improve in the future, as long as the model gets periodically retrained with an augmented

database

## 4.4 Overall performance of fitted model

The overall performance metrics of the fitted model (averaged over a 5-fold cross-validation) at hourly resolution are shown in Figure 8. It appears clearly that the RF models have a lower error (RMSE and scatter) for both liquid and solid precipitation. However on average they tend to overestimate liquid precipitation (positive logBias).

When looking at different precipitation ranges (see Appendix C, it appears that this overestimation only happens at small precipitation intensities ($\leq 2$ mm h$^{-1}$), which still represent the majority of observations (c.f. Figure 2) and for higher values the RF methods show in contrary a slight underestimation with respect to RZC (c.f. Figure C1), indicating that the bias correction is not yet perfect. Note also that the *RF_hpol* model has a consistently poorer performance than the polarimetric model, which could be expected, but still outperforms RZC in terms of RMSE

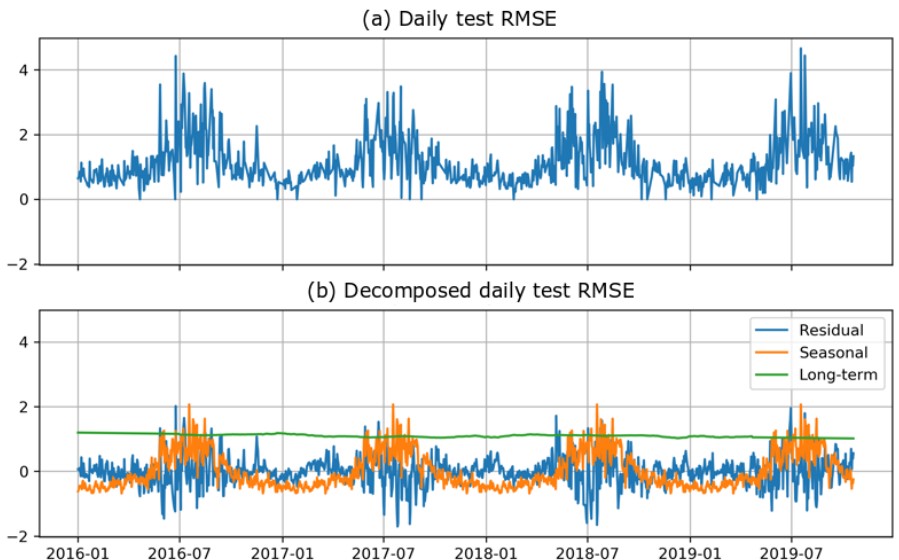

**Figure 7.** (a) Average daily test RMSE obtained from a 5-fold cross-validation, (b) test RMSE decomposed into daily fluctuations, seasonal trend and long-term trend.

and scatter. Though the RF models don't reach the performance of CPC.CV they sometimes get quite close and are less biased for certain precipitation ranges. In general RF_dualpol delivers a performance more similar to CPC.CV than RZC.

An example of prediction versus observations scatter-plots for one iteration of the cross-validation are shown in
Figure 9. It can be seen clearly that for the RF_dualpol method, the spread around the 1:1 line is smaller than for RZC, and there is no visible underestimation trend as can be observed for CPC.CV at higher precipitation intensities.

The performance of the fitted model was also assessed spatially by computing the cross-validation performance metrics separately at every ground station. Figure 10 shows the spatial distribution of the logBias. There is a
clear improvement with the random forest methods, with respect to RZC, as areas of strong underestimation in central-south and central-west Switzerland are not visible anymore. The clear overestimation in Valais (South-west) is however still visible. Performance on other metrics (not displayed) show a clear decrease in scatter and to a lesser extent in RMSE, and a decrease in the ED of RF_dualpol with respect to RZC, although only in central Switzerland.

There is generally a trade-off to be found between the bias (accuracy) and the variance (precision) of a model,
and underestimating models tend to have a smaller error, due to the very asymmetric distribution of precipitation. Figure 11 shows the logBias as a function of the error for all stations. It distinctly shows that RF_dualpol and to a lesser extent RF_hpol are characterized by better trade-off since the logBias is generally closer to 0 and the scatter



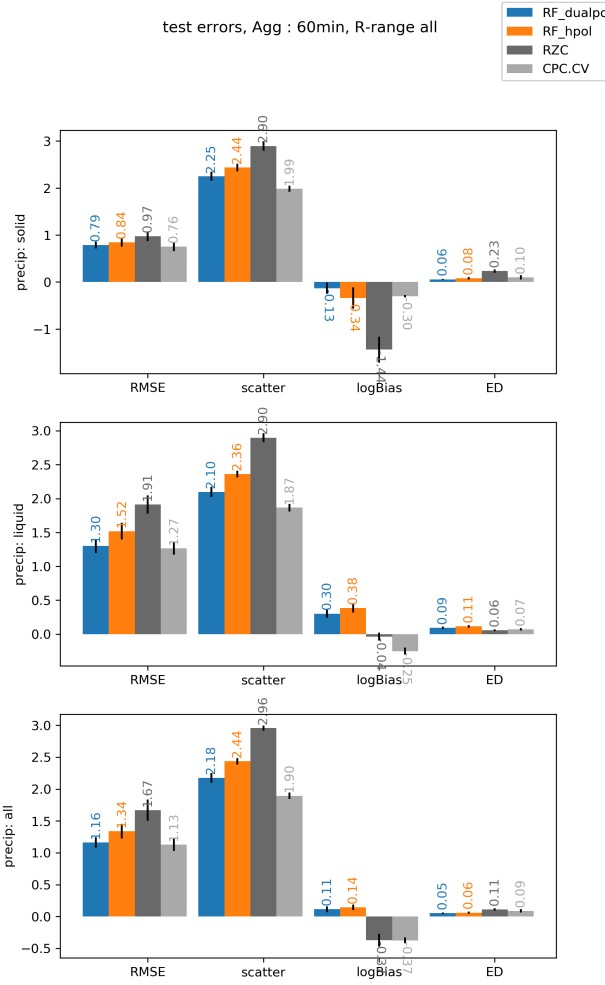

**Figure 8.** Overall performance metrics of the fitted RF models and the references for a 5-fold cross-validation. The small black lines at the center of the bars indicate the standard deviation of the metrics over the 5-fold cross-validation.

is smaller. There is also much less variability between the ground stations, indicating that the model is able to take into account local tendencies.

## 4.5 Error model

Besides a precipitation estimate at every Cartesian pixel, one also needs an estimate of its uncertainty. This uncertainty can be approximated by the error in the cross-validation verification. Figure 12 shows this approximate error as a function of the estimate, as well as possible polynomial fits. It can be seen that the average error tends to be around 50% of the estimate, and the relative error decreases with increasing prediction. The error of the dualpol model is noticeably lower, which emphasizes again that the valuable information brought by the polarimetric vari-

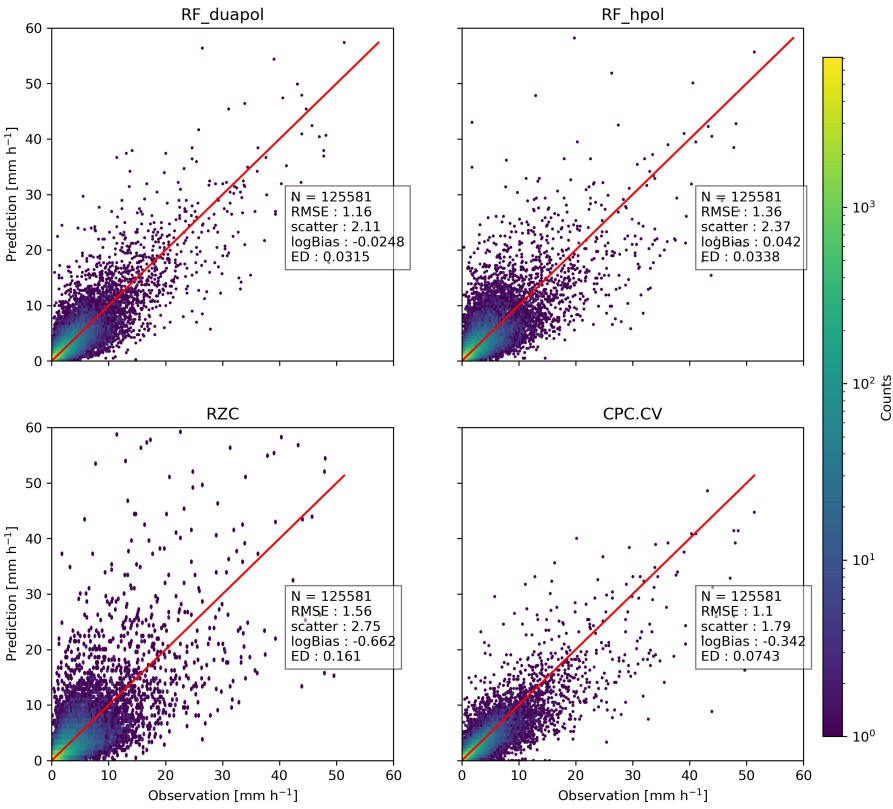

**Figure 9.** Example of prediction versus observations scatter-plots on the test fraction for one cross-validation iteration. The 1:1 line is shown in red.

ables. Though these average errors seem very large they are in fact smaller than the one of RZC, which is shown as a black line.

## 5   Generation of 5 minute QPE maps

The RF QPE method derived from the database, has been adapted to a quasi-operational framework in order to generate 5 min QPE maps at the same temporal and spatial resolution as the RZC product. The main steps remain identical, namely, the spatial averaging of input features to 1 km $^2$, followed by a vertical aggregation to the ground using a logarithmic profile (with $\beta = -0.5$), and finally the use of the trained RF to predict precipitation intensities at the ground. This final estimate is then converted into 256 digital numbers (byte format) using the same lookup table used for the encoding of the RZC product. However, a few modifications were necessary in order to improve the spatial structure of the final RF product and take into account the change in temporal support between the database and the 5-minute QPE maps.

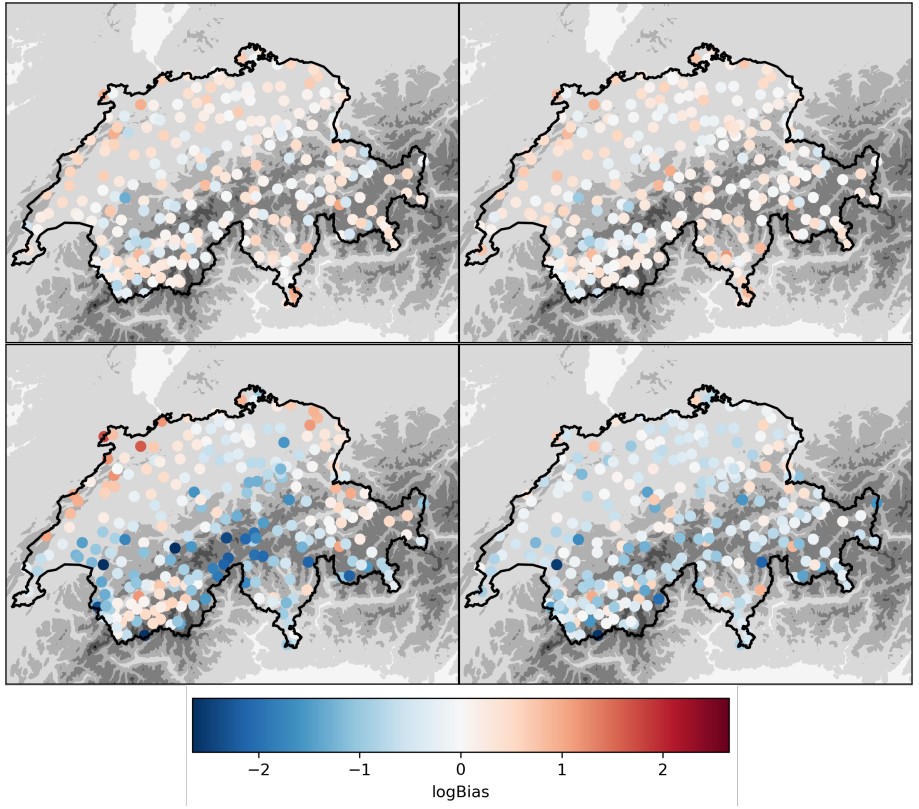

**Figure 10.** Cross-validation logBias at every ground station for RF_dualpol, RF_hpol, RZC and CPC.CV. The displayed geographical domain is the same as in Figure 1. The grey colors in the background illustrate the topography.

## 5.1 Local outlier removal and low-pass filtering

The estimated QPE is generally noisier in space than the standard RZC product. This can be explained by the inherent discretization performed by the RF regression method and by the addition of new weakly intercorrelated input features. To alleviate this issue two operations are performed successively: first a $3 \times 3$ pixels local outlier removal is applied: in every $3 \times 3$ neighbourhood, if the $z$-value (precipitation intensity standardized within its neighbourhood) of a pixel is larger than 3 or lower than -3, its value is replaced by the mean in the neighbourhood. The second step is a low-pass filtering, for which two approaches have been considered:

- A simple 2D Gaussian filtering with a standard deviation of $\sigma = 0.5$ km (0.5 pixel). An explanation regarding this choice of $\sigma$ is given in Appendix D.

- An advection-correction method in which two consecutive 5 minute QPE fields are decomposed into a sequence of 5 fields at a resolution of 1 minute which are then averaged (Appendix 2 of Anagnostou and Krajewski



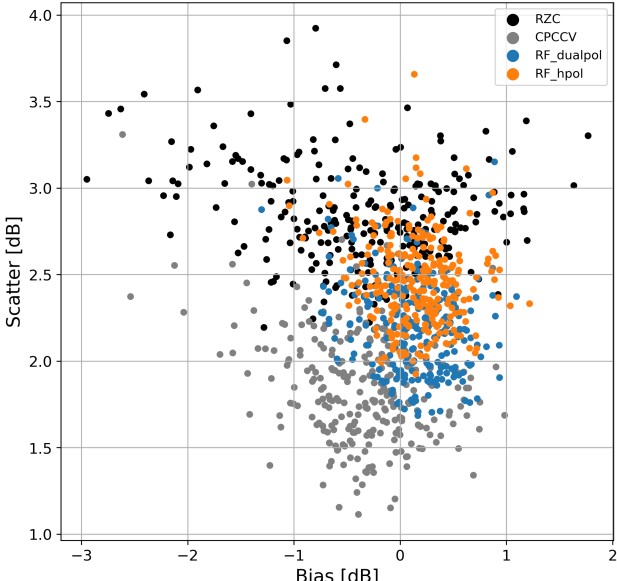

**Figure 11.** logBias versus scatter at every ground station for RF_dualpol, RF_hpol, RZC and CPC.CV. Every dot corresponds to a different station.

(1999)). The Lukas-Kanade optical flow method as implemented in *pysteps* (Pulkkinen et al., 2019) was used to derive the motion vectors of the precipitation fields.

A comparison of these two methods is shown in Figure 13. The advection-corrected field looks much smoother and the intense precipitation cells are larger, although in fact their cores tend to have weaker intensities.

## 5.2    Temporal disaggregation

The RF method has been trained from the database to predict precipitation intensities over a 10 minute period, since it is the smallest available timescale of gauge observations. A disaggregation of the 10 minute estimate delivered by the RF method, is thus necessary in order to match the 5 minute resolution of RZC. In our case we base the disaggregation on the $ZR$ relationship used by MeteoSwiss in its RZC product (Joss et al., 1998):

$$Z_h = 316 R_{ZR}^{1.5} \quad \rightarrow \quad R_{ZR} = \frac{1}{316} Z_h^{\frac{2}{3}} \tag{9}$$

with $Z_h$ in mm$^6$ m$^{-3}$ and $R$ in mm h$^{-1}$, and the subscript ZR indicates an $R$ estimated obtained from this relation.

At a every timestep $t$, the disaggregation is done in the following way:

1. The input features to the RF are computed by taking the average of the two latest 5 minute timesteps.

   $$\overline{\mathbf{X}} = 0.5 \left( \mathbf{X}_t + \mathbf{X}_{t-5m} \right)$$





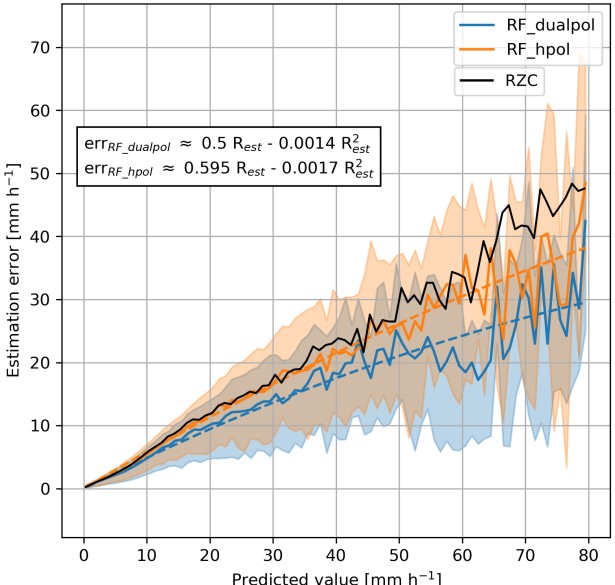

**Figure 12.** Absolute error of the precipitation estimate as a function of the precipitation estimate, evaluated over 5 iterations of 5-fold cross-validation, for the two RF models. The coloured semi-transparent area corresponds to the Q25-Q75 interquantile, the solid lines to the mean, and the dashed line to a polynomial fit, which formulation is shown in the box on the left. The black solid line corresponds to the average error of the RZC model which is shown as a reference. All quantiles and means have been obtained using a discretization on the predicted values with a step of 1 mm h$^{-1}$.

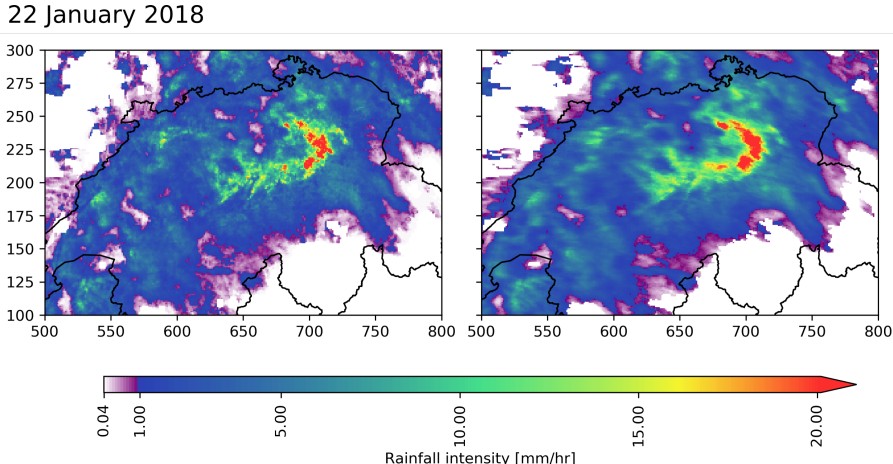

**Figure 13.** Resulting QPE field on a case of widespread snowfall, on the left with Gaussian smoothing ($\sigma = 0.5$ pixels), on the right with advection correction.





2. The 10 minute estimate is computed with the RF model trained on the database

$$R_{\mathrm{RF}}^{(10)} = \mathrm{RF}(\overline{\mathbf{X}})$$

3. The 5 minute RF estimate is obtained by using the fraction of $R_{\mathrm{ZR}}^{(5)}$, estimated from $z_t$ (with Equation 9) to $R_{\mathrm{ZR}}^{(10)}$, estimated from $0.5\,(Z_{h,t} + Z_{h,t-5m})$:

$$R_{\mathrm{RF}}^{(5)} = \frac{R_{\mathrm{ZR}}^{(5)}}{R_{\mathrm{ZR}}^{(10)}} R_{\mathrm{RF}}^{(10)}$$

Figure 14 shows examples of generated QPE fields for two very different precipitation events. The spatial structure of the RF fields looks realistic with no visible radar artifacts such as bright-band (especially for the 22 January event, where mixed phase precipitation is omnipresent), or influence from ground clutter. Despite the Gaussian smoothing the RF fields look somewhat more discontinuous than RZC and particularly CPC (which is naturally smooth due to the kriging procedure). For these two examples, the RF field look like an intermediate stage between RZC and CPC, with weaker and more localized precipitation cores than RZC. This is however not a systematic behavior.

## 5.3   Case-studies

Six precipitation events typical of Switzerland are considered, of which two correspond to widespread winter snow events (warm front on 22 January 2018 and cold front on 29 January 2020), two to summer convection with heavy precipitation (thermal convection on 25 July 2017 and cold-front convection on 6 August 2019), and two to cold front situations in autumn with mainly statiform precipitation (27 October 2018 and 15 October 2019). All events last for 24 hours from midnight to midnight. To ensure an unbiased estimate of the performance, the RF models used for prediction have been trained after filtering out all input data from these 6 days.

The performance of all QPE methods is given in Figure 15 in the form of a color-coded table. It appears that RF_dualpol (with Gaussian smoothing) has a lower RMSE than RZC for 4 out of 6 events (and equal RMSE for the 2 others), a lower scatter than RZC for 5 out of 6 events, a better logBias than RZC for 5 out of 6 events and a lower ED for 4 out of 6 events. In contrast, the performance of RF_dualpol_AC is much poorer and it often overestimates precipitation as can be seen by the larger logBias. In fact this overestimation is limited to small precipitation intensities, which indicates that the smoothing effect is too strong, as the high values in the precipitation cores tend to leak towards the margins of the precipitation system. The performance of the single polarization RF_hpol model is almost systematically worse than RF_dualpol and as such can not really be used as an alternative to RZC. Most likely in the absence of polarimetric information, tbe RF algorithm is not able to overcome the absence of additional VPR and local bias corrections as are applied to the RZC product. When looking at the performance for different precipitation ranges the RF models tends to produce larger precipitation intensities for low observed precipitations, with respect to RZC. In most of the cases, this is rather a good sign, since RZC tends to underestimate at these ranges, but in some cases (15 October 2019 and 6 August 2019) it leads to overestimation

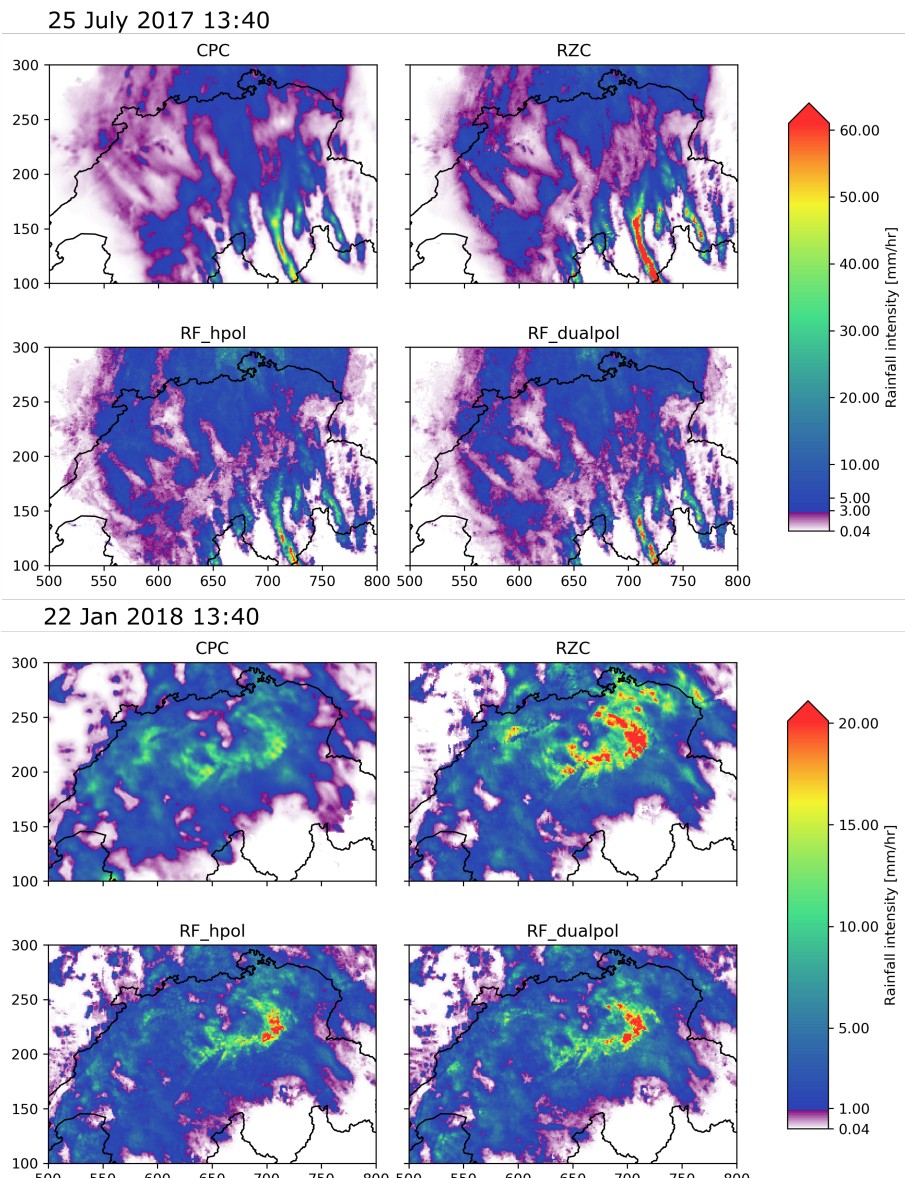

**Figure 14.** Two examples of precipitation fields generated with RZC, CPC, RF_dualpol and RF_hpol (both with Gaussian smoothing): a) heavy snowfall event during a warm front crossing of Switzerland on the 22 January 2018, b) strong convection on the 10 June 2019, during a typical summertime barometric swamp. The colorscale is divided in a linear progression in purple tones for low precipitation and a logarithmic progression from blue to green above a certain threshold.

of weak precipitation. This can be explained by the natural tendency of the RF to overestimate weak responses, which is not totally compensated by the *a-posteriori* bias-correction. Forobserved intense precipitation above 10 mm h$^{-1}$, RZC is clearly underestimating, for all events. RF_dualpol still underestimates but to a much lesser amount,





and the relative decrease in logBias with respect to RZC ranges from 10 to 40% depending on the event. The only event where the performance of RF_dualpol is problematic is the 6 August 2019, where it has a clear tendency to overestimate weak and intermediate (up to 10 mm h$^{-1}$) precipitation). Nevertheless, it is necessary to keep in mind the large difference in the spatial support of the raingauge and the radar, which makes a direct comparison difficult.

As the radar provides an average over a much larger area than the raingauge, it would not be surprising that the estimated values are underestimated with reference to the gauge in case of intense precipitation.

| | 25 Jul 2017 | | | | | 22 Jan 2018 | | | | |
| | RZC | CPC.CV | RF_dualpol | RF_dualpol_AC | RF_hpol | RZC | CPC.CV | RF_dualpol | RF_dualpol_AC | RF_hpol |
| --- | --- | --- | --- | --- | --- | --- | --- | --- | --- | --- |
| RMSE | 0.8 | 0.69 | 0.78 | 0.93 | 0.82 | 1.31 | 0.91 | 1.07 | 1.118 | 1.09 |
| scatter | 2.67 | 2.15 | 2.59 | 2.59 | 2.34 | 2.84 | 1.82 | 2.4 | 2.3 | 2.55 |
| logBias | -1.32 | -0.28 | -0.38 | 0.64 | -0.45 | 0.34 | -0.07 | -0.29 | 0.63 | -0.87 |
| ED | 0.1 | 0.07 | 0.08 | 0.12 | 0.09 | 0.07 | 0.09 | 0.08 | 0.17 | 0.19 |
| | 27 Oct 2018 | | | | | 6 Août 2019 | | | | |
| | RZC | CPC.CV | RF_dualpol | RF_dualpol_AC | RF_hpol | RZC | CPC.CV | RF_dualpol | RF_dualpol_AC | RF_hpol |
| RMSE | 1.13 | 0.74 | 1.01 | 1.07 | 1.09 | 1.2 | 0.83 | 1.2 | 1.33 | 1.66 |
| scatter | 2.62 | 1.39 | 2.38 | 2.32 | 2.63 | 2.41 | 1.82 | 2.11 | 2.65 | 2.07 |
| logBias | -2.03 | -0.11 | -0.39 | 0.5 | -1.07 | 0.89 | -0.02 | 1.61 | 1.52 | 2.63 |
| ED | 0.29 | 0.06 | 0.09 | 0.19 | 0.16 | 0.14 | 0.16 | 0.22 | 0.26 | 0.31 |
| | 15 Oct 2019 | | | | | 29 Jan 2020 | | | | |
| | RZC | CPC.CV | RF_dualpol | RF_dualpol_AC | RF_hpol | RZC | CPC.CV | RF_dualpol | RF_dualpol_AC | RF_hpol |
| RMSE | 1.27 | 0.81 | 1.12 | 1.56 | 1.31 | 0.5 | 0.37 | 0.5 | 0.74 | 0.51 |
| scatter | 2.61 | 1.53 | 2.59 | 2.81 | 2.96 | 2.47 | 2.06 | 2.63 | 2.85 | 2.82 |
| logBias | -1.27 | -0.13 | -0.41 | 0.25 | -0.88 | -1.43 | -0.39 | -0.05 | 0.17 | -0.26 |
| ED | 0.14 | 0.12 | 0.14 | 0.17 | 0.13 | 0.11 | 0.14 | 0.13 | 0.22 | 0.13 |

**Figure 15.** Evaluation scores at hourly resolution for the RZC, CPC.CV, RF_dualpol (with Gaussian smoothing), RF_dualpol_AC (advection-corrected) and RF_hpol (with Gaussian smoothing) methods, for the six events, using the rain gauge measurements from all Swiss stations as reference. Green colors correspond to good relative performance and red colors to poor relative performance.

## 6    Conclusions

In this work we propose a new data-based QPE method for Switzerland, that is able to generate 2D estimates of precipitation intensities over a 1 km$^2$ grid, every 5 minutes, in real time.

The first step of this work involved the creation of large database comprising four years of radar measurements from the 5 operational polarimetric weather radars, simulations from the operational COSMO NWP model, and gauge measurements from the 288 operational rain gauges, aggregated to a common spatial and temporal support.

This database was then used to adjust and train a random forest (RF) algorithm, able to predict the gauge observation at the ground from the radar observations aloft. Compared to other machine learning regression models,

RF has the advantage of being easy to paralellize, very fast for prediction in real-time application, and does never generate non-physical precipitation amounts. Since machine learning methods such as RF typically require the response and the input features to have the same dimensions, the observations aloft are aggregated to the ground using a weighted average that depends exponentially on the altitude of each observation. The relative importance of





each input feature was assessed using a random shuffling scheme and the final choice of features includes 9 features, which are by order of importance: the horizontal reflectivity $Z_H$, the vertical reflectivity $Z_V$, the specific differential phase shift $K_{dp}$, the fraction of observations that come from each of the 5 radars, the copolar correlation coefficient $\rho_{hv}$, the spectral width $S_w$, the temperature from the COSMO model, the static radar visibility and the radar gate altitude. It was observed that the trained RF method has a natural tendency to overestimate weak precipitation and underestimate strong precipitation, which is a well-known behaviour of many machine-learning methods. This tendency can be alleviated to a large extent with an a-posteriori bias-correction method, that relies on a fit between observed precipitation and predicted precipitation. The final model includes the following hyper-parameters: (1) the slope in the exponential altitude weighting of the input features, (2) the number of trees, (3) the maximum depth of the trees, (4) the number of variables randomly chosen at each split and (5) the type of a-posteriori bias-correction. By running a 5-fold cross-validation for every parameter combination, it was observed that the performance differs greatly as a function of the precipitation intensity range and the type of metrics that is used. There is thus no single best-choice but a good trade-off could be found with a multi-criterion decision scheme. The learning curve computed for this fine-tuned algorithm reveals that the training data is sufficient but that the required data is quite large in regard of the small number of input features. This result can be explained by the intrinsic loss of information due to the aggregation of the gauge data to the spatial and temporal support of the gauge and the inherent noisiness of some of the radar variables.

Comparison of this new algorithm with RZC, the current single-polarization QPE product of MeteoSwiss, reveals that it decreases significantly the estimation error and bias in most areas of Switzerland. This is particularly true in the central Alpine regions, where RZC tends to underestimate. Nevertheless, even though the bias-correction method solves to a large extent the issue of underestimating heavy precipitation, for which the RF QPE is generally better than RZC, the RF algorithm still has a visible tendency to overestimate weak precipitation. When training an RF QPE without the polarimetric information, the performance is generally much poorer and worse or comparable to RZC. Some effort was also invested in the computation of an error model, which allows to estimate the error associated the predicted precipitation intensities. This model reveals that the polarimetric information reduces the error clearly when compared with RZC, or RF without polarimetry.

To be able to provide 5 minute precipitation maps, the algorithm was adapted with a disaggregation scheme that predicts 5 minute estimates from 10 minute averaged input features (which is the temporal support of gauge observations and hence the one used for training the RF algorithm). This scheme relies on the ZR-relationship used in the operational RZC model. It was observed that the resulting QPE fields could display locally sharp discontinuities, which are not visible in RZC, which is generally smoother. These discontinuities can be explained by the use of new additional input variables in the QPE as well as the discretization performed by the RF regression. To improve the spatial structure of the output, the QPE scheme was complemented with a $3 \times 3$ local outlier removal filter and a Gaussian low-pass filter with $\sigma = 0.5$ km. A series of six case-studies for typical precipitation events over





Switzerland, reveals that the generated precipitation look realistic and gives a better performance than RZC for 5 out of 6 events.

This new RF QPE method has proven to deliver promising results and has the advantage of replacing many of the a-posteriori corrections required by RZC (global and local bias corrections, VPR correction) by one single fast

estimation step. Further work is required to improve its capability to predict weak precipitation intensities, which might require a more sophisticated aggregation scheme of radar observations aloft. This QPE algorithm offers vast perspectives for operational real-time applications, indeed it is fast (less than a minute of computation every 5 minutes), and because of its simple structure (ensemble of decision trees) the RF regressor can easily be used in the operational MeteoSwiss framework, provided the radar variables are transformed into input features accordingly. In

parallel, the database will periodically be updated with newly acquired data and the RF regressor will be retrained. Ultimately this new RF QPE should serve as input to the CPC algorithm which provides the best possible QPE estimate over Switzerland by merging radar QPE and gauge data (Sideris et al., 2014).

*Code and data availability.* . Data and code are available on request by contacting the authors.

*Acknowledgements.* The authors would like to thank Ioannis Sideris for all help regarding the CPC algorithm and its products.
The authors are also grateful to Marco Boscacci and Jordi Figueras i Ventura for the advice regarding all MeteoSwiss operational products.

## Appendix A: Computation of $K_{dp}$ and $A_h$

To create a database of four years of radar data, more than 10 million radar PPI scans have to be processed (5 radars, 20 elevations every 5 minutes). Because of this, it is computationally impossible to use an advanced $K_{dp}$

estimation method, such as a Kalman filter method (Schneebeli et al., 2014) or a Gaussian-mixture regression (Wen et al., 2019). Hence a simple method is used in this study which is also used in Wolfensberger et al. (2018) and is similar to Timothy et al. (1999). The raw total differential phase shift $\Psi_{dp}$ is first corrected for the system offset and then filtered with a moving median filter to give an estimate of the total differential phase shift on propagation $\Phi_{dp}$. To estimate $K_{dp}$, which is half of the slope of $\Phi_{dp}$, a moving linear regression is used, where the slope is estimated

in a moving window. For sake of consistency the same window length is used both for the median filtering of the phase and the linear regression. Tests showed that the best results are obtained by using a large window of 6 km.

Concerning $A_h$, we use the ZPHI method (Testud et al., 2000), in liquid phase only. The COSMO temperature is used to identify the height of the freezing level. Attenuation is neglected in the solid phase (i.e. $A_h$ is always zero





above the freezing level). In the liquid phase, we use constant values of $\gamma = 0.08$ and $b = 0.64884$, as provided by default in the Pyart package (Helmus and Collis, 2016).

## Appendix B: Cross-validation results

Figures B1 and B2 show the RMSE for all precipitation intensities resp. only high precipitation intensities. It
clearly shows that even at large precipitation intensities using no bias correction (first column) gives quite large errors. These plots also show that even when considering only RMSE it is difficult to find a good trade-off in the choice of hyperparameters.

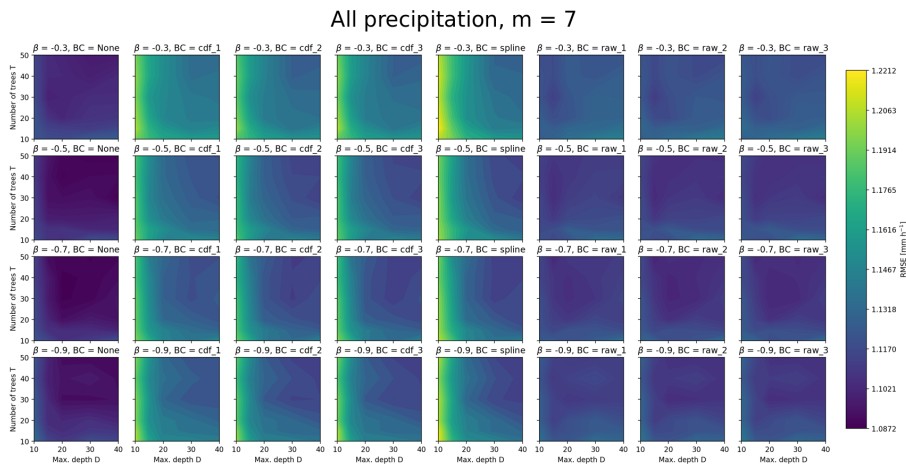

**Figure B1.** Overall RMSE, for all observations as a function of number of trees, maximum depth, $\beta$ parameter and bias-correction method. The maximum number of randomly chosen variables at each split is here set to 7.

## Appendix C: Overall performance at different precipitation ranges

Figure C1 shows the overal cross-validation performance of RZC, CPC.CV, RF_dualpol and RF_hpol over the
whole database, separated by precipitation phase and intensity.

## Appendix D: Choice of $\sigma$ in the Gaussian smoothing

The optimal $\sigma$ value in the Gaussian smoothing of the 2D QPE fields has been chosen by analyzing the overall performance in terms of RMSE and linear bias during the six representative precipitation events as a function of the value of $\sigma$. Intuitively if the smoothing is too intense, the natural tendency of the RF to overestimate weak
precipitation and underestimate strong precipitation could be worsened. The results, displayed in Figure D1, show

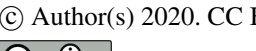


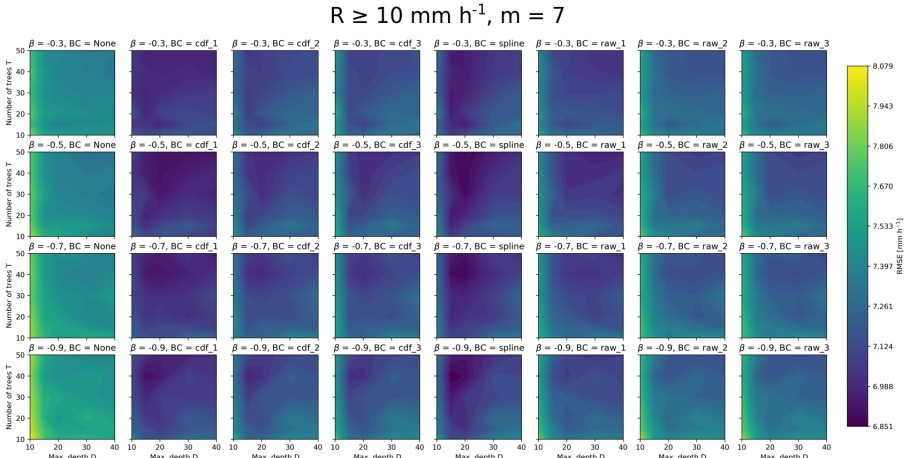

**Figure B2.** Overall RMSE for observations $\geq 10$ mm h$^{-1}$ as a function of number of trees, maximum depth, $\beta$ parameter and bias-correction method. The maximum number of randomly chosen variables at each split is here set to 7.

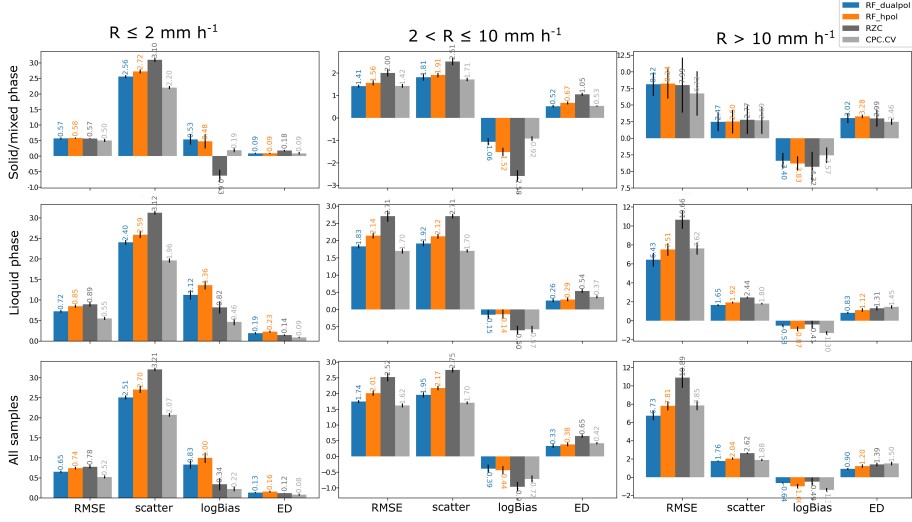

**Figure C1.** 5-fold cross-validation performance of RZC, CPC.CV, RF_dualpol and RF_hpol for different observed precipitation ranges for liquid and solid/mixed precipitation and for both together (which includes all additional observations from stations where no temperature measurement is available). The y-scale is different for all precipitation ranges, since the scores are significantly different.

that increasing $\sigma$ leads to a decrease of the RMSE but also to a sharp increase (in magnitude) of the bias. A value of $\sigma = 0.5$ km seems to be a good trade-off since it leads to a comparatively low increase in bias for a comparatively large decrease in RMSE.

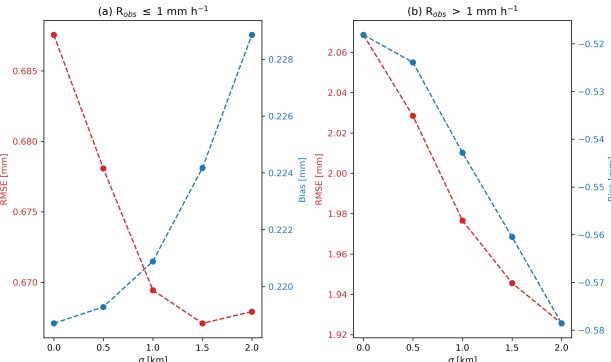

**Figure D1.** Hourly RMSE and linear bias (average of estimated vs observed value), as a function of the value of $\sigma$, (a) for low intensities, (b) for higher intensities.

*Author contributions.* DW designed and implemented the QPE algorithm, performed all experiments detailed in this work, and wrote the manuscript. AB, MG, MB and UG contributed to the design and discussion of the work, as well as to the writing of the manuscript.

*Competing interests.* The authors declare that they have no conflict of interest.



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
