# Peer review of "RainForest: A random forest algorithm for quantitative precipitation estimation over Swizerland"

_Atmospheric Measurement Techniques, 2020_

## Referee Comment (RC1) · Anonymous Referee #1 · 9 Nov 2020

The work deal with quantitative precipitation estimations (QPEs) in complex horography. Both weather radar based estimations and ground measurements suffer from several limitations. Hence, combining both source of observations is mandatory but challenging. The present study proposes a new technique based on training on random forest regression to learn QPE from large database. One advantage of this algorithm is the relative lo cost in term of computation, making it appealing for real time applications. The algorithm 's evaluation is carried out by the authors assessing errors and bias. Moreover, six cases. characterized by different precipitation types (stratiform, convective). The work is scientifically interesting, clearly exposed with strictness. An attractive aspect of the paper is its look towards an operational application for real time

[Figure]

QPE. Only minor revisions are needed:

pag. 1 line 3 relations -> relationships pag. 3 line 8 However -> However, pag. 3 line 21 Finally -> Finally, pag. 3 line 27 Thus -> Thus,

Figure 5 (b) it is unclear, several labels overlap.

About "minute" the authors should choose between x-minute or x minutes . In the text expressions like "5 minute" or "10 minute" are frequent.

---

## Referee Comment (RC2) · Irene Crisologo (Referee) · 13 Nov 2020

**Review of "RainForest: A random forest algorithm for quantitative precipitation estimation over Swizerland"**

By Irene Crisologo

**Summary:**

The paper presents a new method for Quantitative Precipitation Estimation, by using the Random Forest algorithm to convert radar variables into precipitation estimates. The authors created multiple products using the Random Forest algorithm, by using different inputs such as dual-pol variables, horizontal-polarization variables, and bias-corrected data. Their results suggest that Random forest show promising results for QPE.

Overall the paper is well written, the methods are presented well, and the results are conclusive. Just a few clarifications are needed, and some minor things edited for consistency.

**General Comments**

1. The inclusion of computational methods, a written algorithm, and citation of programming libraries are greatly appreciated.
2. The paper (understandably) uses several acronyms, and as they are defined in various locations in the text, it might help readability if an additional table defining acronyms is added. I understand that this is not a common thing to add to papers, so the authors may consider this optional.
3. The efficacy of the RF QPE is compared to a Z-R based QPE. Do dual-pol based QPE products exist that can also be used as a reference?
4. Table 1 is not referenced in the text.
5. What is the reason for selecting only 6 cases from 4 years of data?
6. The final products that the authors compared in the end could be presented in a more reader-friendly manner. The "RF_dualpol_AC" product for example is first mentioned in Page 24, and when I searched for the definition within the text, it is only defined in the caption of Figure 15 on Page 26.
7. Random forest is the main method and core idea of the paper, yet little about it has been mentioned in the introduction. Maybe a bit more information on Random Forest can be added in the introduction.
8. I would also like to suggest for the authors to check their figures against a colorblindness simulator for color schemes and choices that might not be very accessible.
9. I appreciate the vector figures, zooming in does not blur the image. However, the need to zoom in could be eliminated by having a consistent, readable font size throughout the manuscript. Some figures have font sizes that are too small. Ideally, the font sizes in the figure matches the font size of the manuscript body, or only a little smaller.

**Specific comments**

1. Page 1, line 13: "… that appears less smooth then the original…" change *then* to *than*
2. Page 3, line 26: COSMO is mentioned for the first time, but not defined. Please add a sentence briefly describing/introducing COSMO, as it is referred to in the later parts of the manuscript.
3. Page 5 line 10: Ott Pluvio footnote is missing.

4.  Page 7, Table 2: The shorthand VIS could be added to the "Visibility" row of this table, since VIS and "Visibility" is used interchangeable throughout the manuscript.
5.  Page 10, Figure 3: Maybe a monochrome colorscale could be used for the density plot? The green part of the colorscale slightly interferes with the green and blue lines in the plots. The colorbar is also missing.
6.  Page 10, Figure 3: The legend entries are not defined anywhere. What do "raw_deg1", "cdf_deg1", and "cdf_spline" mean? Does the density plot in Panel (b) correspond to the green line in Panel (a)?
7.  Page 12, line 14: What are the stations mentioned and where are they located? A bit more context about the mentioned stations could help.
8.  Page 14, Figure 5: Much discussion is presented about Figure 5, it would help if Figure 5 is more readable and refined. The labels of the pie chart overlap, which makes it hard to read. NOISE_H and Zh has the same color of pie slice. Zv and RVEL also seem to be the same color. Some labels are also not defined anywhere, e.g. FRAC_RADAR. The variables also have inconsistent use throughout the manuscript, e.g. $Noise_H$ vs NOISE_H, RVEL vs $R_{vel}$
9.  Page 16: There is a table but the table number and caption are missing. This table should also be referenced in the text.
10. Page 18, Figure 7: Is there a difference between "residual" in the legend and "daily" in the caption? Or do they refer to the same thing?
11. Page 19, Figure 8: Some labels overlap each other or the standard deviation indicator.
12. Page 23, Figure 13: What do the x- and y-axis represent?
13. Page 24, line 22: "RF_dualpol_AC" is used here before being defined in Figure 15 caption.
14. Page 26, Figure 15. The authors defined green colors as good performance, red colors as poor performance. However, there are multiple shades of greens and reds, as well as yellows. A legend or colorscale showing the thresholds used for the different colors and shades could be useful.
15. Page 26, Figure 15: The CPC.CV column seems to be the best performing of all methods, as it is mostly green. Discussion about this is missing in the text.

**Technical corrections**

1.  Page 1, title: Swizerland -> Switzerland
2.  Page 1, line 18: "Still providing" -> "Still, providing"
3.  Page 6, line 2: Plane -> Plan
4.  Page 10, Figure 3 caption: "Panel (a) shows ab example…" *ab* → *an*
5.  Page 10, Figure 3 caption: Kindly clarify this part of the sentence: "…which are estimate the observations as a function…"

**Very minor and somewhat nitpicky corrections**

(I've recently read some books about data visualization and designing plots, and found this review to be a good opportunity practice identifying the little things that could improve a figure. However, they do tend to be nitpicky, so feel free to skip these suggestions!)

1. Figure 3
    a. The red 1:1 line could extend a bit more so that it goes from corner to corner of the plot, like it does in Figure 4.
    b. The colorbar is missing.
    c. The 1:1 line eliminates the need for the gridlines.
    d. The figure can be made bigger.
2. Figure 4
    a. The font size of the labels should be increased for readability.
    b. The aspect ratio could be set to 1:1 for each subplot
    c. The legend box alignment can be improved (my suggestion would be to place it on the top right, where it interferes least with the data)
    d. The figure can be made bigger.
3. Figure 5
    a. If possible, pie chart slices should appear in decreasing size, so that the order of slices (whether CW or CCW), has some logical sequence.
4. Figure 7
    a. The assignment of colors for the (a) Daily test RMSE and the Residual line in (b) Decomposed daily test RMSE makes it seem like they are the same variable.
5. Figure 9
    a. The red 1:1 line could extend a bit more so that it goes from corner to corner of the plot.
    b. The legend box alignment can be improved.
6. Figure 12
    a. The grid lines can be eliminated.

---

## Author Comment (AC1) · 22 Dec 2020

Please see Section 1 of the attached PDF

Please also note the supplement to this comment:
https://amt.copernicus.org/preprints/amt-2020-284/amt-2020-284-AC1-supplement.pdf

---

## Author Comment (AC2) · 22 Dec 2020

**Review RainForest**

daniel.wolfensberger

December 2020

**1  Reviewer 1**

We are thankful to reviewer 1 for agreeing to review the article and for his helpful recommendations. We have improved the article in the following way:

- We have fixed the missing commas after However, Finally and Thus (see modified version of the article), as well as the term relations by relationships in some sentences.

- Regarding Fig.5, indeed as was also pointed out by Reviewer 2, this figure needed some improvement. It has hence been edited by replacing the labels with names that agree with the text of the paper, by solving the issue of overlapping labels and by choosing different colors for all slices of the pie-charts.

- We have now chosen to use the notation x-minute throughout the article, when referring to a certain quantity (for example 10-minute QPE maps).

**2  Reviewer 2**

We are thankful to reviewer 2 for reviewing the article and for the many useful suggestions that are proposed. We will address here every point that was raised by the reviewer.

**2.1  General Comments**

1. Thank you.

2. We thought about this point and tried to add a table in the appendix. However, this table ended up being quite short (only a few items) since it included specific acronyms only (CPC, RZC, RF_dualpol, RF). Indeed, we think that acronyms such as QPE, VPR, RMSE, and others are fairly common in the radar meteorology field and should not necessarily be summarized in a table. In the end, since this table was so short it was decided not to keep it.

3. Unfortunately, there is no dual-polarization QPE product available at MeteoSwiss at the moment. A model could be implemented from the literature but it would require to be adapted and calibrated to the Swiss radar system, which would require an amount of work similar to the development of this QPE RF. Unfortunately it could not be run retroactively over the four years of data in the archive as that would be too computationally expensive. It could thus only be used for the event-based comparison (Section 5.). Before experimenting with RF, we had implemented a simple KDP estimation method and had run it for a couple of events, but the performance was very poor. Even if it is non-polarimetric, RZC has been specifically designed for the Swiss radar configuration and topography (VPR correction) and has been fine-tuned over many years. As such we think that it is a fairer competitor to RF than an uncalibrated polarimetric QPE model taken from the literature.

4. Thank you for pointing this out. We have added the following line in the paper: *Table 1 summarizes the differences in spatial and temporal support between all different data sources used in this work.*

5. The overall scores reported in Section 4. consider all 4 years of observations. The goal of the event based analysis in Section 5. is to verify if this overall performance gain is also valid for individual events. We decided to choose only 6 different events corresponding to very different weather situations, because it makes it easier to get a specific overview. Indeed Figure 15 is already very dense with 6 events only. Considering more events would make it difficult to assess the performance separately for each event and assessing the average performance for all events, would not be very relevant as this would be somewhat redundant with Section 3.

6. We have added an explicit reference to RF_dualpol_AC in Section 5.1. *In the following the term RF_dualpol_AC will be used for the RF product obtained with dual-polarization inputs and smoothed with the advection-correction method.*

7. We have added some short description of random forests in the paragraph 5 of the introduction.

8. Thank you for this advice, I must admit I tend to forget this issue. All plots have been checked with a colorblind simulator[1]. Fortunately most figures were readable because green and red colors are usually not present together in the figures. However, Figures 4 and 15 were problematic. The colormaps for these figures have thus been replaced by more colorblind friendly ones.

9. We agree that some fonts were too small to be readable. We have increased the font size in the most problematic figures, in particular Fig.1, 4, 5 8 and 9.

**2.2 Specific comments**

1. Thank you, this has been fixed

2. We have removed this first reference to COSMO and replaced it by *(from point measurements to large area numerical weather prediction fields)*. Few lines later the COSMO model is introduced a bit more clearly: *For numerical prediction, MeteoSwiss runs the COSMO Model which is a mesoscale limited area model that is operated and developed by several weather services in Europe (e.g. Switzerland, Italy, Germany, Poland, Romania, and Russia) ....*

3. This is not a footnote, but the name of the model. Indeed Ott uses an exponent for the "'2', which I agree is not very convenient when written in a text: https://www.ott.com/products/accessories-109/ott-pluvio2-weighing-rain-gauge-963/.

4. We have changed the term visibility to VIS in Table 2 and Figure 5, to be more consistent.

5. This figure has been edited by using a monochrome colormap for the density plot and by using colorblind friendly colors for the lines. The colorbar has been added.

6. The names in the legend have been modified to agree with the Table in p.16 (note that this table was previously unreferenced in the article but is now Table 3) as well as the text in Section 3.2. We have also emphasized the fact that the green fit in the left panel is used to get the density plot in the right panel.

7. We have added the full name of the stations which should make their location more explicit for people who are familiar with the geography of the Alps. We have also given some context on their location (mountain summits or passes).

8. This figure has been modified to make the pie-charts larger and to make sure no colors are duplicated and no labels overlap. The labels have also been modified to agree with Table 2. COncerning $FRAC_{RADAR}$, it is now refered to as $Frac_{rad}$ throughout the text and it is explained in Section 3. We have also added one sentence in the caption of this Figure to make this clearer. *The importance of the feature $Frac_{rad}$ is the sum of the importance of the fraction of every single radar (see Section 3.3).*

9. This has been fixed, the table has now a caption and a label and is refered in the text.

10. They are different terms. Daily implies that all displayed values in the first panel are daily average RMSE values. Then this signal of daily average RMSE values is decomposed with an algorithm into a long term trend, a seasonal trend and residuals (i.e. fluctuations that are neither seasonal nor a long-term trend). Residual refers thus to fluctuations in the daily RMSE that have neither seasonal trend nor long-term trend. It was quite confusing in the text, because the term daily was used for fluctuations as well. We have made this clearer in the caption and in the text above the figure.

11. We have removed overlaps between labels and standard deviation indicators. The figure should be more readable now. Note that the new figures shows very tiny deviations from the values in the previous figure due to the fact that the cross-validation procedure was repeated to generate a new plot and it is stochastic by nature.

12. We have added x an y-axis labels which make it clear that these are Swiss CH1903 coordinates in km scale.

13. Besides the added definition of RF_dualpol_AC in Section 5.1, we have added a definition in the text in line 22: *In contrast, the performance of RF_dualpol_AC, the RF dual-polarization QPE with a-posteriori advection-correction (Section 5.1), is much poorer and it often overestimates.*

14. We have changed the colormap of this plot to a transition from red (worst performance) to blue (best performance through white. This makes it easier to read and is more colorblind friendly. An important thing is that the colormap is defined separately for every score and every event. The worst performing method has the darkest red and the best performing method the darkest blue, all others have colors that are interpolated linearly in between. Indeed it is difficult to defined a global colormap that would be valid for all events, since the scores are very much dependent on the intensity of the event. We have added a colorbar at the bottom of the plot to make it clearer.
* * *
[1]https://www.color-blindness.com/coblis-color-blindness-simulator/

15. We have added the following sentence at the beginning of this paragraph: *The best-performing QPE is almost systematically CPC.CV. Since this method uses also ground measurements, the comparison with pure radar products such as RZC and the RF QPE, is not fair, but the performance of CPC.CV can be considered as an asymptotic ideal performance, to which the best-possible radar QPE should tend.*.

**2.3   Specific comments**

1. What a shame...this has been corrected.

2. Corrected

3. Corrected

4. Corrected

5. This was changed to :   *which try to estimate the observed value as a function of raw predicted values.*

**2.4   Very minor corrections**

Due to the relatively short time available for the review, some of the points here could not be tackled, but we have tried to address as many as possible.

1. Besides the modifications explained in point 6 of 2.2, we have also extended the 1:1 lines to the limits of the figure, removed the grid and increased the size of the figure from one column to two columns in the text, as was suggested.

2. We have changed the aspect ratio to 1:1 for each subplot, have modified the legend box to reduce the overlap with the density plot and have increased the font size to make it larger. Unfortunately we cannot change the size of the figure as AMT uses standard sizes (8.3 cm for one column, 12 cm for two columns).

3.

4. Color of the line in the top panel has been changed to black instead of blue.

5. We have extended the 1:1 line to the corners and have moved the legend box so it doesn't overlap the subplots borders.

6.